# MALDI-IMS combined with shotgun proteomics identify and localize new factors in male infertility

Shibojyoti Lahiri[1], Wasim Aftab[1,4,*], Lena Walenta[2,*], Leena Strauss[3], Matti Poutanen[3], Artur Mayerhofer[2], Axel Imhof[1]

Spermatogenesis is a complex multi-step process involving intricate interactions between different cell types in the male testis. Disruption of these interactions results in infertility. Combination of shotgun tissue proteomics with MALDI imaging mass spectrometry is markedly potent in revealing topological maps of molecular processes within tissues. Here, we use a combinatorial approach on a characterized mouse model of hormone induced male infertility to uncover misregulated pathways. Comparative testicular proteome of wild-type and mice overexpressing human P450 aromatase (AROM+) with pathologically increased estrogen levels unravels gross dysregulation of spermatogenesis and emergence of pro-inflammatory pathways in AROM+ testis. In situ MS allowed us to localize misregulated proteins/peptides to defined regions within the testis. Results suggest that infertility is associated with substantial loss of proteomic heterogeneity, which define distinct stages of seminiferous tubuli in healthy animals. Importantly, considerable loss of mitochondrial factors, proteins associated with late stages of spermatogenesis and steroidogenic factors characterize AROM+ mice. Thus, the novel proteomic approach pinpoints in unprecedented ways the disruption of normal processes in testis and provides a signature for male infertility.

## Introduction

Infertility is a major health condition that affects around 15% of couples of reproductive age worldwide. About a third of these occurrences are due to male infertility. During the recent decades, male reproductive health has declined in many parts of the world contributing, cumulatively, to about 65% of all infertility cases worldwide (Li et al, 2006; Singh & Singh, 2017). A variety of chromosomal anomalies like Klinefelter syndrome (Gravholt et al, 2018), Y-chromosome deletions (Song et al, 2016), and mutations in spermatogenic genes (Bracke et al, 2018) have been attributed to reduced fertility in men. Apart from these, a number of anatomical and physiological features have been associated with infertility in men, but precise causative molecular mechanisms has remained elusive so far. This has resulted in a substantial percentage of male infertility cases to be classified as idiopathic (Singh & Singh, 2017) and, hence, subject to only empirical treatments. Systematic understanding of spermatogenesis at the molecular level is therefore imperative in attempts to tackle the aforesaid condition.

Spermatogenesis is a process that results in the production of fully differentiated male gametes. It involves a series of intricate interactions between different cell types and molecules within the testis. A number of genetic, epigenetic, and environmental factors have been shown to affect spermatogenesis leading to reproductive failure in men (Neto et al, 2016). In most cases, the testis is the principal organ giving rise to abnormalities leading to male infertility. High levels of intratesticular testosterone are required to produce spermatozoa and impaired sperm production can result from increased circulating estrogen levels (Ciaccio, 1978, Jones et al, 1978; Kalla et al, 1980). To simulate such a hormonal imbalance a transgenic mouse model overexpressing the human P450 aromatase (AROM+) gene has been established (Li et al, 2001; Li et al, 2006). These mice have an increased E2/T ratio and develop a progressive male infertility phenotype. The disease model has so far been only characterized on a histological and phenotypic level resulting in the identification of a limited number of affected pathways (Li et al, 2001; Li et al, 2006; Strauss et al, 2009). Detailed molecular description of the system is still lacking, which results in a limited understanding of pathways involved. Large-scale -omics techniques allow a much deeper molecular understanding of the pathways and proteins involved in the phenotype. So far, most proteomic studies have focussed on sperm or seminal fluids (Selvam & Agarwal, 2018). Only very few proteomic studies of the testis have been performed, which were in addition severely limited

[1]Biomedical Center, Protein Analysis Unit, Faculty of Medicine, Ludwig-Maximilians-Universität München, Planegg-Martinsried, Germany    [2]Biomedical Center, Cell Biology-Anatomy III, Ludwig-Maximilians-Universität München, Planegg-Martinsried, Germany    [3]Institute of Biomedicine, Research Centre for Integrative Physiology and Pharmacology and Turku Center for Disease Modeling, University of Turku, Turku, Finland    [4]Graduate School for Quantitative Biosciences (QBM), Ludwig-Maximilians-Universität Munich, Munich, Germany

Correspondence: imhof@lmu.de
*Wasim Aftab and Lena Walenta contributed equally to this work

in terms of the number of proteins identified. This, along with the lack of spatiotemporal information, have led to very inadequate understanding of the functional network of these proteins in spermatogenesis (Liu et al, 2013; MacLeod & Varmuza, 2013). Present-day shotgun proteomic studies of complex organs can identify a large number of proteins. However, they inherently bear the disadvantage of loss of spatial information of the identified proteins. To overcome this inherent problem for a better understanding of spermatogenesis and male infertility, we first performed a deep shotgun proteomic study of testis from 11-mo-old AROM+ mice and compared them with wild-type controls. The information about the identified peptides gathered this way was then used to learn about their in situ distribution using matrix assisted laser desorption/ionisation imaging mass spectrometry (MALDI-IMS). The aforesaid exhaustive studies in combination with a newly developed data analysis tool described here provide an in-depth understanding of molecular processes affected in the testis during hormone induced and inflammation-associated male infertility.

## Results

### Characterization of cellular pathways and proteins upon alteration of E2/T ratio in testis

Mice overexpressing aromatase (AROM+) are known to exhibit symptoms of human male infertility (Li et al, 2006). The mice show cryptorchidism, Leydig cell hyperplasia and a general lack of reproductive fitness. As the processes dysregulated in these animals are not entirely clear, we performed an extensive proteomic analysis of testis tissues from WT and AROM+ mice (Fig 1A) at the age of 11 mo.

A total of 4,327 proteins were identified, among which, 476 and 624 proteins were only detected in the WT and AROM+, respectively (Fig 1A Venn diagram, Table S1). A subsequent statistical analysis of the shared 3,227 proteins revealed a further significant enrichment (>4-fold in at least two out of three animals) of 27 proteins in the healthy mice as compared with 75 proteins in the AROM+ system (Fig 1B, brown filled circles, Table S1). To identify the dysregulated cellular processes upon overexpression of aromatase, we analysed the molecular pathways to which these proteins contribute. Results reveal that regulation of androgen biosynthesis and metabolism of steroid hormones are the most highly enriched processes in WT mice (Fig 1C and Table S2). Proteins involved in these pathways such as Cyp17a1 and Hsd17b3 (Table S2) have previously been shown to be severely reduced in AROM+ testis (Strauss et al, 2009). Apart from proteins involved in the abovementioned pathways, a number of other proteins were significantly enriched in WT mice. The presence of proteins like the testis-specific isoform of Estrogen sulfotransferase, sperm mitochondrial-associated cysteine-rich protein, axonemal dynein heavy chain 12 and 17, phosphoglycerate kinase 2 and tubulin polyglutamylase TTLL5, testis-specific cytochrome C, and male-enhanced antigen 1 in WT but not AROM+ mice (Table S1) suggest a lack of late-stage spermatogenesis (Nayernia et al, 2002; Ohinata et al, 2002; HumanProteinAtlas) because of aromatase overexpression.

We also find a number of biological processes (BPs) up-regulated in the testes of AROM+ animals. So are, for example, ECM organization and disassembly highly enriched in AROM+ animals suggesting induction of pro-inflammatory pathways upon alteration of E2/T ratio (Fig 1D and Table S3). The induction of a substantial immune response upon AROM overexpression is also consistent with the increased detection of factors involved in the regulation of the complement cascade (Fig 1D), neutrophil degranulation, and platelet aggregation. A pathway analysis of the proteins enriched in AROM+ mice shows that the composition of the ECM is likely mediated by the regulation of collagen and the metabolism of glycosaminoglycans, two major components of the ECM (Fig 1D and Table S3).

### Molecular pathways severely affected upon increased E2/T ratio

The proteins detected exclusively in either WT or AROM+ testis also provide valuable insights into the most predominant BPs dysregulated by an elevated E2/T ratio. Pathway analyses show that proteins exclusively detected in WT form three distinct clusters (Fig 2A–C). The first cluster represents various pathways related to transcription initiation and elongation (Fig 2A and Table S4), whereas the second cluster reveals processes involved in transcription termination and RNA maturation (Fig 2B and Table S4). This is in agreement with the substantial loss of germ cell differentiation and proliferation, known to be transcriptionally highly active. The third cluster (Fig 2C and Table S4) reveals mitochondrial processes to be predominant in the WT. Absence of these pathways in AROM+ along with its fertility defects suggest that higher estrogen levels impair mitochondrial function, potentially being largely responsible for the impaired steroidogenesis in the AROM+ testis. In addition, AROM+ animals demonstrate up-regulation of immune response factors (Fig 2D and Table S5), platelet activation, degranulation, and other platelet dependant regulatory processes along with neutrophil degranulation that suggest the triggering of severe inflammatory and immunomodulatory activities in the testis of 11-mo-old AROM+ mice. Furthermore, higher levels of proteins associated with ER and Golgi function is observed suggesting prevalence of cellular stress in AROM+ (Table S5).

### In situ proteomic map parallels the morphological change in AROM+ testis

Increase in estrogen levels has been found to associate with cell type specific effects in male testis. Prominent fibrosis has also been observed in the interstitial spaces and tubular walls of the testis in aged AROM+ mice (Li et al, 2006). To generate a molecular map that reflects the morphological changes in AROM+ testis and to get a better understanding of the distribution of particular proteins and peptides within the tissue, we generated peptide maps of the testis from MALDI-IMS measurements by digesting WT and AROM+ mice testis sections with trypsin in situ. An unbiased hierarchical clustering of the acquired IMS data enabled us to distinguish different regions of the testis solely based on spectral features (Fig S1).

Results show two distinct types of primary clusters in the segmentation map of both WT and AROM+ represented by the brown and dark blue patches in Fig 3A and B. The brown cluster represents

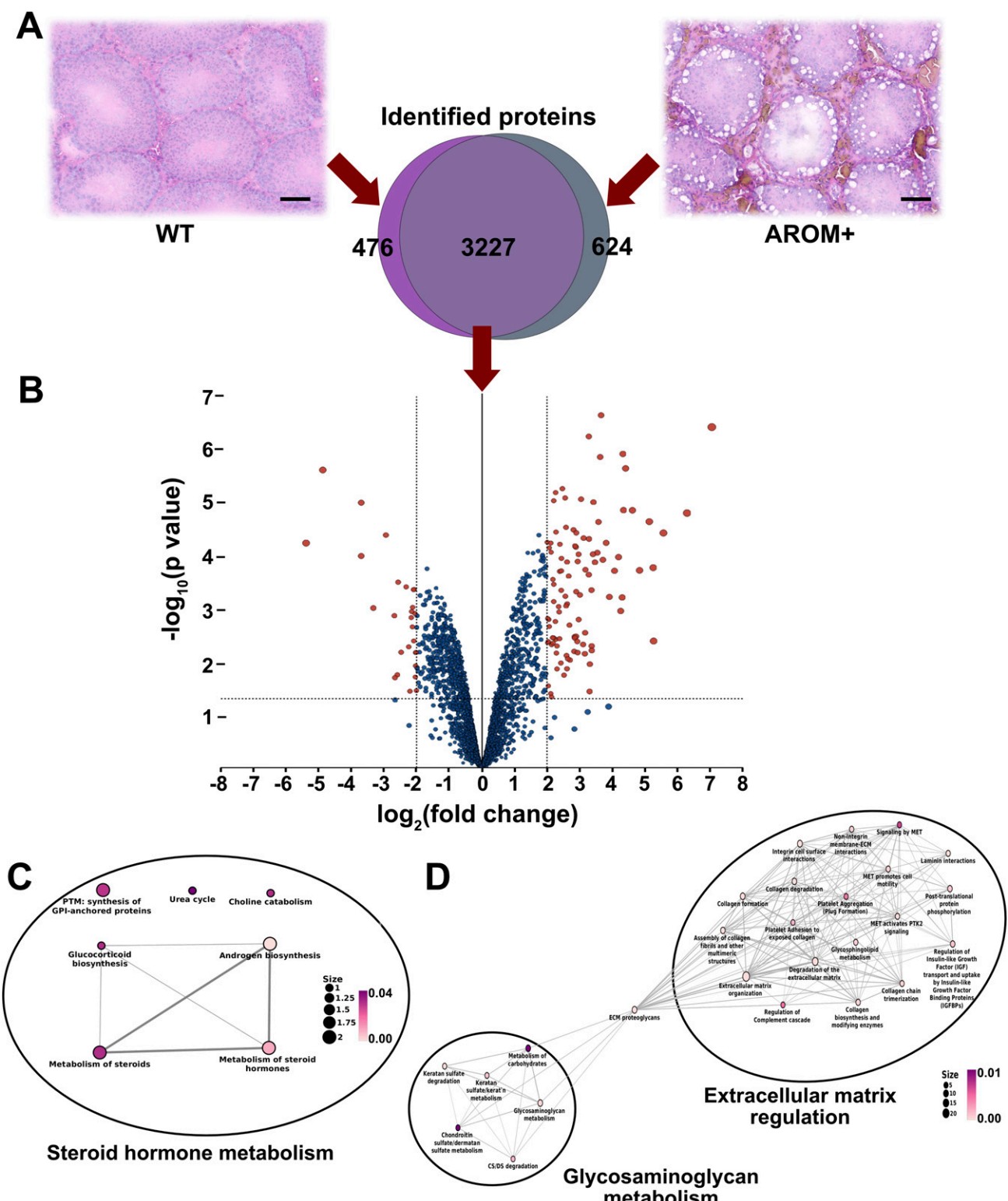

**Figure 1. Proteomic analysis of WT and AROM+ testes.**
**(A)** The upper panel shows HE stained tissue sections of WT and AROM+ testis. Scale bar—100 *µm*. Venn diagram in the lower panel shows that 3,227 proteins were identified in both WT and AROM+. In addition, 476 and 624 proteins were detected exclusively in WT and AROM+, respectively. **(B)** Volcano plot generated from relative abundance of the shared 3,227 proteins after statistical analysis by LIMMA moderated *t* test. The filled brown circles represent the significantly enriched proteins in both WT (negative fold change) and AROM+ (positive fold change). **(C)** Statistically significant molecular pathways generated by enrichment analysis using hypergeometric distribution model represented by the WT enriched proteins (Table S2). **(D)** Statistically significant molecular pathways using enrichment analysis (pathways having the

peptides distinct to certain testicular seminiferous tubules (Figs 3A and S2), which are greatly reduced in AROM+ (Figs 3B and S2) derived tissue (~52% in WT as compared with 9.5% in AROM+). This peptide signature is in line with the observed overall disruption of tubular morphology in AROM+ testis (Li et al, 2006).

Detailed examination of the segmentation maps enabled us to further assign distinct spectral features to tubules (lumina and peritubular regions) or interstitial spaces (Fig 3C; red, brown, green, dark blue, orange, and royal blue, respectively, Fig S3). The tubular clusters along with their finer structures are less prominent in case of AROM+ (Figs 3D and S3). Besides a general lack of a spectral diversity among the testis from AROM+ mice, we identified an additional interstitial cluster in these animals compared with a WT control (Fig 3E and F yellow-green, Fig S4).

### Identification of proteins in situ

To identify the most likely parent proteins of the peptides representing different spectral clusters, we developed a bioinformatic pipeline that combines IMS and liquid chromatography mass spectrometry (LC–MS/MS) measurements using a maximum likelihood strategy (Fig 4).

The complexity of the tissue section used in MALDI-IMS results in multiple overlapping isotopic envelopes derived from the peptides present at a particular region within the tissue (Fig 4A, panel: MALDI-IMS). This makes the deisotoping for any given peptide, which is required for all search algorithms extremely challenging and has been a major hurdle in MALDI-IMS based proteomics. To overcome this issue we took advantage of an unbiased hierarchical clustering of the entire imaging dataset (Fig 4A, panel: hierarchical clustering), which results in a segregation of the spectra on the entire tissue section based on their spatial distribution pattern. As we expect all isotopes of a given peptide to show the same spatial distribution, this strategy allowed us to separate overlapping isotopic envelopes with a very high accuracy (Fig S5). In mouse testis this enabled us to unambiguously assign a regional distribution to 369 unique deisotoped peptides from our MALDI-IMS spectra. To identify these peptides, we compared this mono-isotopic list with the peptides identified by LC–MS measurements of serial sections (Fig 4B). However, because of the moderate mass accuracy (±0.1 Da, Fig S6 and Table S12) for peptide measurements in the current setup, this comparison often resulted in an ambiguous assignment of each m/z value to several peptides identified in the LC–MS analysis (Fig 4C). These apparently isobaric peptides could therefore lead to biased pathway analysis. To correctly assign such peptides to a particular protein, we devised a novel scoring method termed as "most likely peptide (MLP) score." A particular m/z value from IMS was assigned to that identified peptide (from LC–MS analysis), which had the highest MLP score. Using this strategy we could identify masses discriminating the clusters from each other (Fig 4D).

The scoring could identify the spermatid associated Axonemal dynein heavy chain 12 (Dnah12) and sperm motility enabling Sperm mitochondrial-associated cysteine-rich protein (Smcp) among others in the largest cluster (red) of tubules of WT mice (Fig 4D and Table S6). The next largest cluster (brown) shares the distribution pattern of the abovementioned proteins but in addition also contains peptides corresponding to argininosuccinate synthetase (Ass1) and testis-specific gene A8 protein (Tsga8) among the significantly enriched proteins in WT (Fig 4D and Table S7). As most of these proteins were shown to play a role in final stages of spermatogenesis and fertilization, we assume that the cluster corresponds to tubular stages IX-XI of spermatogenesis. Either of the clusters are barely detected in AROM+ mice (Fig 3D) supporting the hypothesis that aromatase overexpression interferes with late stages of spermatogenesis (https://github.com/wasimaftab/IMS_Shotgun).

The most prominent peptide defining the exclusive interstitial cluster of the AROM+ testis (Fig 4D) is derived from the ECM protein mimecan (Ogn), a component of the keratan sulphate degradation pathway. Interestingly, mimecan has been shown to be differentially regulated in men suffering from idiopathic germ cell aplasia (Alfano et al, 2019). Owing to the pixel based statistical specificity and sensitivity of the discriminative mass finding algorithm of SCiLS Lab program, it was not possible to get other distinctive masses for this cluster. However, MLP scoring of a manually curated mass list showing specific interstitial distribution in AROM+ led to the identification of different collagen isoforms and prolargin, which are both involved in ECM assembly and regulation (Fig 4D and Table S8).

### Loss of inter-tubular heterogeneity and tubule associated biological pathways in AROM+

A notable feature of the IMS measurements is a substantial inter-tubular heterogeneity (Fig 5A; red, brown, and deep blue groups referred to as tubuli A, B, and C, respectively, from here on) in the WT tissue suggesting differential protein composition of the tubules at the same cross-section. Because different tubules in the same histological section of mouse testis represent different stages of spermatogenesis, it is likely that there are proteomic signatures correlating to these stages. Absence of this heterogeneity in AROM+ tubules reflects disruption of tubular stages and hence severely impaired spermatogenesis (Fig 5A).

A complete gene ontology (GO) analysis of the proteins identified in the MALDI-IMS dataset by applying MLP scoring strategy shows that the two largest tubular clusters (A and B tubuli) are enriched for proteins involved in processes that are almost exclusively involved in late stages of spermatogenesis and fertilization (binding of sperm to zona pellucida, sperm–egg recognition, cell–cell recognition, single fertilization, and cilium assembly and organization) (Fig 5B and C and Tables S9 and S10). In addition to these shared proteins/pathways, tubuli A are rich in mitochondrial protein(s)

lowest 25 P-values generated using hypergeometric distribution model are displayed here) represented by the AROM+ enriched proteins (Table S3) (data from three mice for each group WT and AROM+ was considered for the statistical analyses). The colour code is representative of the statistical significance and is displayed according to the adjusted P-values generated using hypergeometric distribution model (as indicated by the magenta-white gradient bar). The size of the nodes in the network (scale represented by numbered black solid circles) are representative of the number of genes involved in a particular pathway within the network.

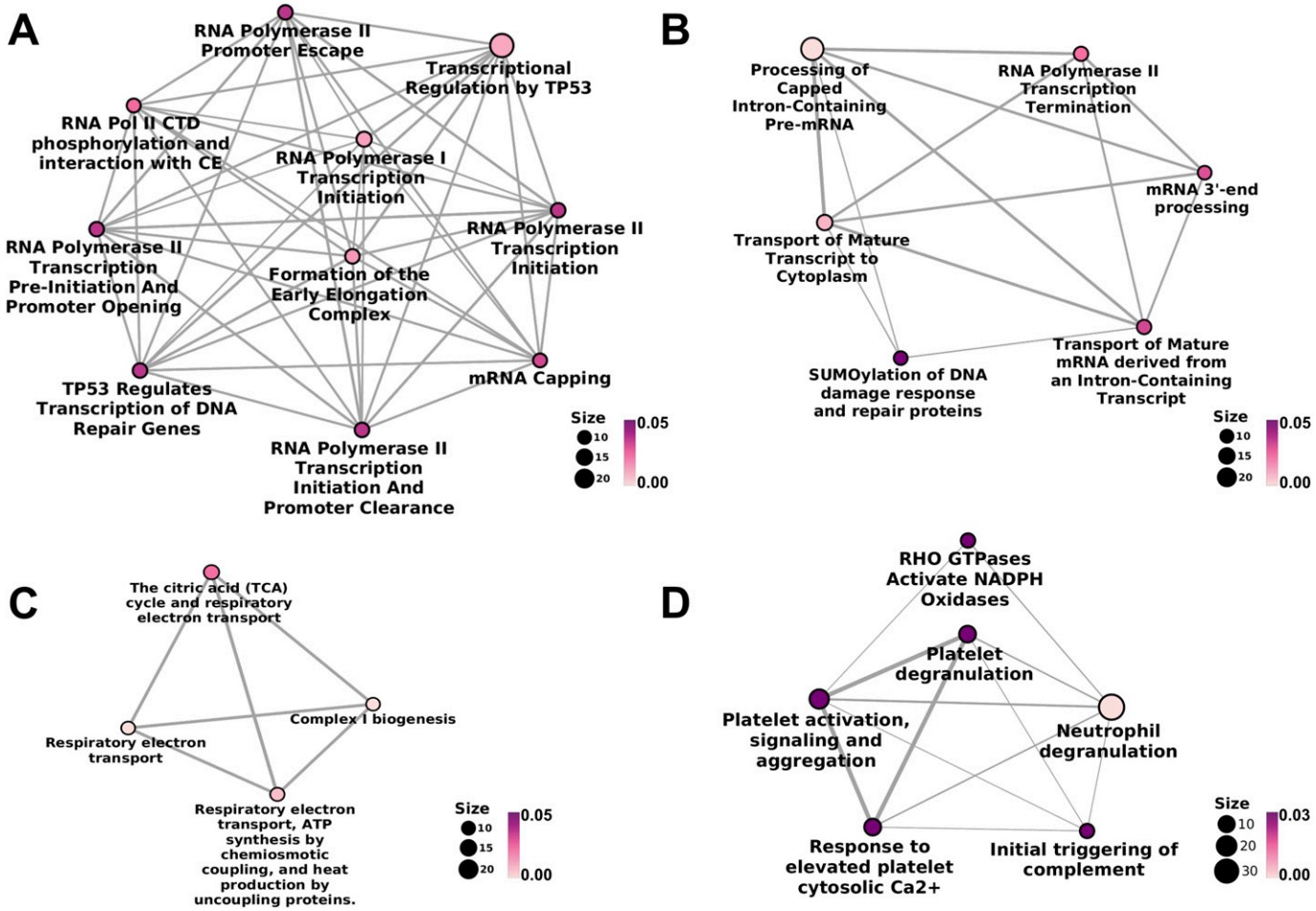

**Figure 2. Pathways exclusive to WT and AROM+ testes.**
**(A)** Significant molecular pathways associated with early phases of transcription, detected exclusively in WT. **(B)** Pathways associated with later stages of transcription that are detected only in the testes of healthy animals. **(C)** Mitochondrial pathways observed solely in WT. **(D)** Inflammatory pathways observed only in 11 mo old AROM+ mice, which are otherwise not observed in healthy conditions (n = 3 mice for each group WT and AROM+). The colour code is representative of the statistical significance and is displayed according to the adjusted *P*-values generated using hypergeometric distribution model (as indicated by the magenta-white gradient bar). The size of the nodes (scale represented by numbered black solid circles) is representative of the number of genes involved in a particular pathway within the network.

(that represent the tricarboxylic acid [TCA] cycle) and factors involved in male gamete formation (Table S9), whereas tubuli B have higher levels of proteins regulating ciliar organization and cell projection assembly (Table S10). Finally, tubuli C are characterized by ER chaperone Hspa5 and its co-chaperone Dnajb11, which together with Wdr35 suggests ciliary transport/trafficking as a distinct feature of this set of tubules (Fig 5D and Table S11). Proteins defining these clusters are observed to be enriched in WT as compared with AROM+ in the bulk proteomic measurements (Fig 5: insets in each panel). Near absence of clusters A and B in AROM+ mice is in line with the data showing that tubules involved in late spermatogenesis and fertilization are severely disrupted in this model of male infertility.

Therefore, it is observed that the healthy animals are characterized by sets of tubules with varied functions representing the complexity that is present and required for healthy spermatogenesis in testis. Impaired spermatogenesis, especially the late stages, in AROM+ leads to a severe reduction of this proteomic

complexity, implying the necessity of region-specific molecular profiles in male gamete formation.

### Validation of MLP scoring based distribution of altered proteins in the testis

IMS measurements and identification of proteins based on our strategy of mass comparison and MLP scoring could demonstrate the molecular changes leading to the difference in morphology between WT and AROM+. However, for direct confirmation of the results of our approach, we proceeded to affirm the distribution pattern of some up- and down-regulated proteins in AROM+ by antibody based experimental techniques.

In our measurements it was observed that Mimecan, a protein component of the ECM was up-regulated in AROM+ (Tables S1 and S8). Peptide distribution of Mimecan (m/z: 763.5, according to MLP scoring) showed an interstitial distribution pattern in the AROM+ derived tissue (Fig 6A; upper panel, IMS). Immunohistochemical (IHC) studies showed identical results in paraffin

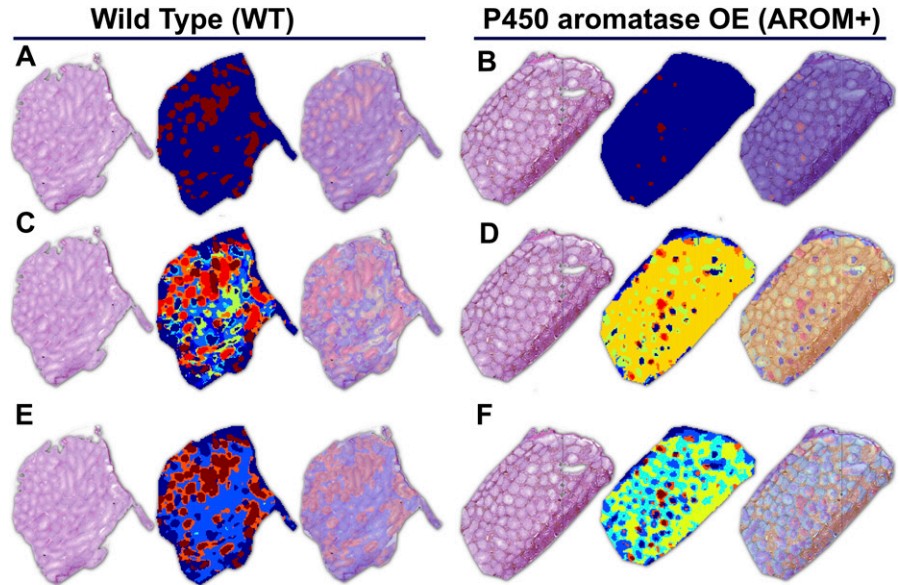

**Figure 3. Proteomic map of WT and AROM+ testes.**
**(A, B)** Initial unbiased hierarchical clustering of IMS peptide data from generated from whole tissue sections of (A) WT and (B) AROM+ testis (dendrograms: Fig S2) at 25 µm spatial resolution. **(C, D)** Further clustering results for the (C) WT and (D) AROM+ tissues (Dendrograms: Fig S3). In case of WT, four different sets of tubules are represented by the red, brown, green and deep blue clusters, respectively. The orange cluster represents peritubular regions and the royal blue cluster characterizes the interstitial spaces. **(E, F)** A further level of clustering of (E) WT and (F) AROM+ tissues (dendrogram: Fig S4) shows an exclusive cluster (yellow-green) that is localized to the interstitial regions of AROM+ testis. Scale bar—2 mm.

embedded Bouin fixed testis tissue sections from mice of similar age (Fig 6B; lower panel, IHC), thereby validating our observation and confirming the involvement of hitherto unidentified Mimecan in the AROM+ phenotype. To further substantiate, we stained testis cross-sections with a collagen I antibody, which is observed to be significantly up-regulated in the AROM+ (LC–MS data, Table S1). The staining shows an interstitial distribution (Fig 6B; lower panel, IHC) corroborating our estimation by IMS measurements (Fig 6B; upper panel, IMS; m/z—929.6, according to MLP scoring). It is observed that although collagen I is present in both WT and AROM+, it is present in AROM+ in higher amounts. Most of the down-regulated proteins in AROM+ showed a tubular distribution supporting the hypothesis of an overall disruption of tubular function in AROM+. Many of the identified proteins using MLP scoring (i.e., Smcp, Vdac2, and Hspa2) have been shown to have a tubular distribution pattern in human testes cross-sections using specific antibodies (HumanProteinAtlas; Uhlén et al, 2015). To validate our novel findings, we stained the WT and AROM+ tissues for Cct6a (T-complex protein 1 subunit zeta), which is a part of the chaperonin-containing T-complex (TRiC) and has been shown to be involved in sperm–oocyte interaction (Dun et al, 2011). Identical to our mass spectrometric findings (Fig 6C; upper panel, IMS) results show strong tubular staining in the WT tissue and no detectable protein in AROM+ (Fig 6C; lower panel, IHC).

## Discussion

Aromatase overexpressing (AROM+) mice are infertile and testicular changes closely resemble the ones observed in infertile men (Li et al, 2001). This progressive testicular dysfunction results in an almost complete lack of spermatogenesis in older mice (Li et al, 2006) at which stage the model provides a good phenocopy of human male infertility. It was the goal of this study to better characterize the molecular phenotype of these mice at this stage rather than analysing causes and consequences of aromatase overexpression. We therefore performed an extensive proteomic analysis of the testes of 11-mo-old AROM+ and WT animals only combining bulk high-resolution shotgun proteomics with spatially resolving MALDI-IMS. However, thanks to the establishment of the measurement pipeline described in this manuscript, such a course of molecular changes during earlier time points of development of infertility as well as a detailed analysis of human biopsies will be possible in future studies.

The rationale for combining these two techniques is the lack of information on regional expression patterns in complex tissues when performing shotgun proteomics analyses and the difficulties of identifying individual peptides using MALDI-IMS. In addition, the method can overcome the issue that proteins not significantly enriched in bulk proteomes tend to get omitted from further investigation thereby leading to an underestimation of their role in the diseased phenotype. This is, for example, the case for a number of spermatogenesis related proteins such as Odf2, Vdac2, Clgn, Akap4, Dnajc3, and Tdrp which are abundant in tubuli A and B and Dnajb11 and Hspa5 specific for tubuli C in the system used here. All of the proteins mentioned above are not significantly altered in the bulk analysis but clearly show a very different regional expression when investigated by MALDI-IMS suggesting that they play an important role for the pathology studied. It is important to mention that our strategy is based on proteomic differences between two conditions and is currently not efficient to characterize clusters based on a single condition alone or clusters that are similar in proportion between two groups (e.g., the green cluster in Fig 5A). To do this, a complementary and much more labour intensive strategy of applying

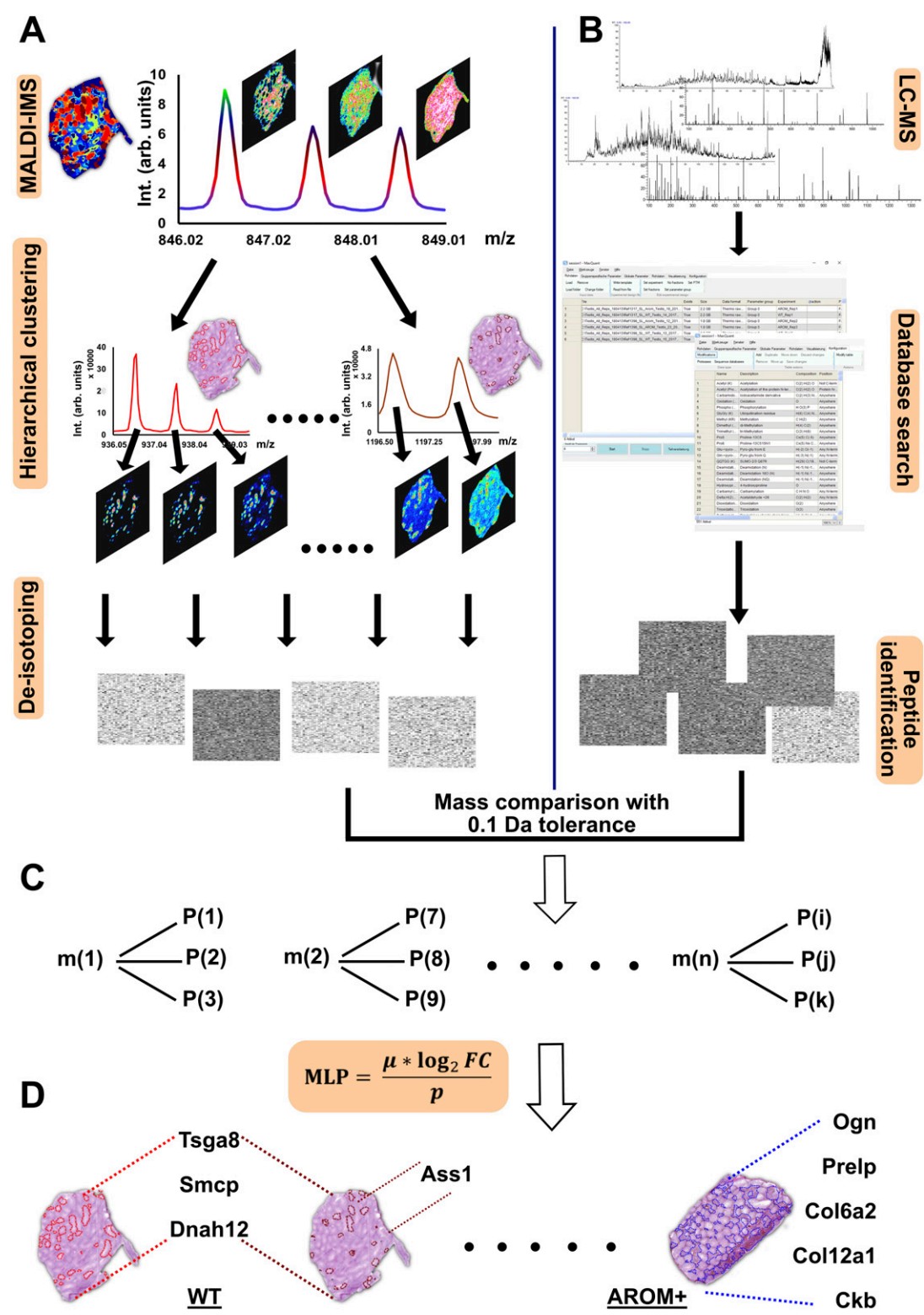

**Figure 4. Protein identification in situ with maximum likelihood.**
**(A)** Imaging mass spectrometry (IMS) measurements lead to an overlap of isotope envelopes of different peptides in the same mass range (*Panel: MALDI-IMS*). This is indicated by different distribution patterns of the concerned m/z peaks. Therefore, to segregate the spectra, we have used unbiased hierarchical clustering (SCiLS Lab 2016b) of IMS data from the entire tissue (*Panel: Hierarchical clustering*) that led to efficient deisotoping of the IMS spectra and create a mono-isotopic list (*Panel: De-isotoping*). The grey rectangles represent the mono-isotopic mass list obtained from each cluster. **(B)** LC–MS/MS measurements were performed on tissue following the section used for IMS and peptides/proteins were identified following spectra analysis and database search (MaxQuant 1.6.0.16). The grey rectangles represent the

laser capture microdissection (LCM) to the regions defined by such clusters would be required (Clair et al, 2016; Zhu et al, 2018; Eckert et al, 2019). LCM can of course also be used to characterize clusters that are different between the two tissues where it will also provide an immediate identification of selected peptides. However, because of the largely increased analysis time of LCM-MS (each region of interest requires the sectioning, sample preparation and a 90-min LC–MS gradient), we think that-upon the assignment of specific peptides by our pipeline, MALDI-IMS has a clear advantage in a more clinical setting where multiple patient samples need to be analysed in a reasonable timeframe.

The combination of bulk LC–MS with MALDI-IMS and the localized identification of proteins defining particular in situ clusters allowed us to identify a substantial alteration of proteomic composition of seminiferous tubules between WT and AROM+ mice pointing towards a severe impairment of spermatogenesis in AROM+. We also observe that the interstitial space in AROM+ testis is characterized by a unique peptide cluster, which is consistent with the previous observation of an interstitial fibrosis in AROM+ mice.

An interesting corollary of the combined analysis is the fact that it can be used to identify and molecularly characterize distinct sets of seminiferous tubules in the WT testis based on their proteomic composition. The set designated as A tubuli is defined by peptides derived from a series of proteins involved in fertilization related processes. Similar to the A tubuli, the B tubuli are characterized by factors associated with spermatogenesis, male gamete formation and associated metabolic processes. In addition to the shared functionalities, our identification strategy could also distinguish between these two sets of tubuli, highlighting the proteomic diversity of seminiferous tubules required for proper spermatogenesis. The C tubuli are enriched in endoplasmic reticulum chaperone BiP and its co-chaperone DnaJ homolog subfamily B member 11, which are part of the ER associated unfolded protein response in cells (Chen & Brandizzi, 2013). The higher number of this type of tubuli in AROM+ compared with the type A and B tubuli suggests that AROM+ mice elicit an unfolded protein response during sperm maturation along with a triggering of pro-inflammatory pathways.

Our exhaustive proteomic approach not only corroborates previous findings of involvement of Leydig cells in AROM+ phenotype but also provides a potential molecular explanation for the physiological impact of the aromatase induced increase in E2 levels. A striking feature of the differential testis proteome of AROM+ is the loss of various mitochondrial pathways and associated proteins in AROM+ mice (Fig 2 and Table S4). Energised and respiring mitochondria play an important regulatory role in LH-dependent steroidogenesis

(Allen et al, 2006), which is likely to be depleted in AROM+ mice despite normal levels of LH. This effect of increased aromatase levels on mitochondria is also consistent with the observation that mitochondria in Leydig cells of AROM+ are significantly enlarged (Strauss et al, 2009). In addition to the loss of mitochondrial factors involved in steroidogenesis, we also detect significantly lower levels of estrogen sulfotransferase in AROM+ mice. This enzyme is abundantly expressed in Leydig cells and catalyses the formation of an estrogen sulfo-conjugate, thereby reducing the levels of active non-conjugated estrogen in the male testis (Qian et al, 2001). These novel findings reveal that sustainable high levels of E2/T ratio in AROM+ mice is not only due to an increased E2 synthesis but also due to a lack of endogenous androgen synthesis and an impaired sequestration of active estrogen by enzymatic conjugation.

In summary, the combination of two complementary proteomic methods allows us to identify and localize proteins and molecular pathways involved in establishing a diseased phenotype. Our novel and unique bioinformatic strategy to identify peptides from IMS measurements based on a combination with bulk proteome analysis of healthy and diseased tissue is not only able to characterize the molecular processes that result in hormone induced male infertility but opens up new avenues for further exploration of cellular processes in situ.

# Materials and Methods

### Animals

Adult male mice were housed at Helmholtz Zentrum Munich under specific pathogen free conditions in GM500 cages, including individually ventilated caging systems (IVC System Green Line; Tecniplast), which are operated with positive pressure. The mice were transferred to new cages with forceps in Laminar Flow Class II changing stations weekly; they were fed with an irradiated standard rodent high-energy breeding diet (Altromin 1314; Altromin Spezial futter GmbH & Co. KG) and had access ad libitum to semi-demineralized filtered (0.2 mm) water. The light cycle was adjusted to a 12/12 h light/dark cycle; room temperature was regulated to 22 ± 1°C and relative humidity to 55% ± 5%. Husbandry conditions were adjusted to the experimental requirements in specified modules. Sentinels (outbred 8-wk-old male SPF Swiss mice) were housed on a mixture (50:50) of new bedding material and a mixture of soiled bedding from all cages of this IVC rack and their health was monitored by on-site examination of certified laboratories according to the FELASA recommendations. Animal handling was in accordance with German and European guidelines for use of animals for research

identified peptide dataset. **(C)** Upon mass comparison, each m/z value (m(1), m(2)...m(n)) from the IMS mono-isotopic list was, in many cases, annotated to multiple peptides identified in LC–MS/MS measurements (P(1), P(2)...P(k)) because of moderate mass accuracy (0.1 Da) of the current IMS setup. To clear the aforesaid ambiguity, we developed a scoring system (MLP scoring) that enabled to identify the parent proteins with maximum likelihood. **(D)** Distribution of identified proteins (proteins having highest MLP scores) in the WT and AROM+ testis. Proteins involved in late stage spermatogenesis (that were among the significantly altered proteins in LC–MS measurements also) were identified in the seminiferous tubules of WT (Table S7), whereas proteins involved in ECM regulation were found in the interstitia of AROM+ testis (Table S8).

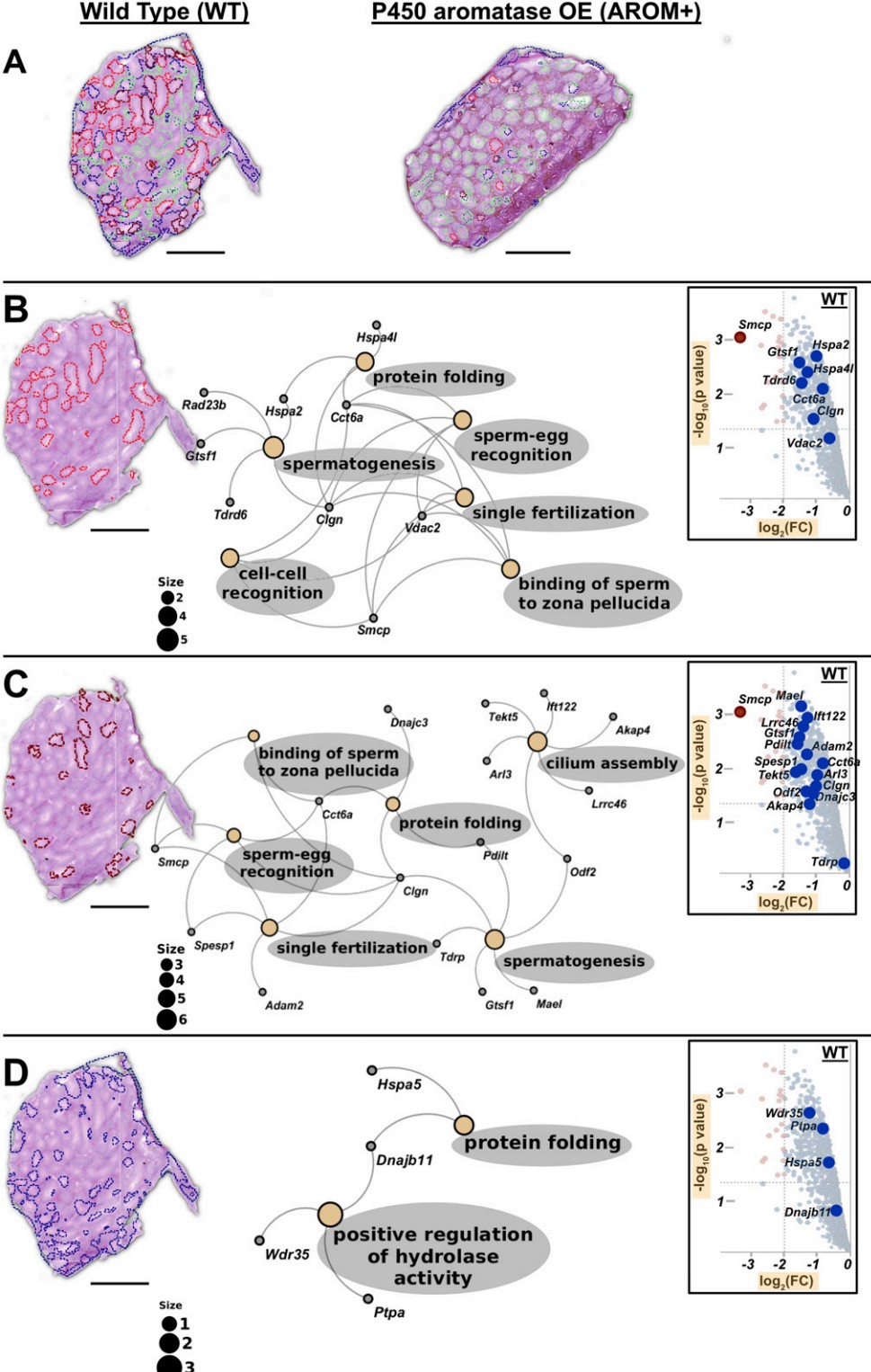

**Figure 5. Inter-tubular heterogeneity and associated processes.**
**(A)** WT testis showing inter-tubular heterogeneity (red, brown, green, and deep blue) based on their protein composition. AROM+ testis (on the right side) does not exhibit this proteomic heterogeneity. **(B)** Gene ontology (GO) term analysis of first set of tubules (red, tubuli A) based on the proteins identified there by MLP scoring. **(C)** GO term analysis of the second set of tubules (brown, tubuli B) based on the proteins identified there by MLP scoring. **(D)** GO term analysis of the third set of tubules (blue, tubuli C) based on the proteins identified there by MLP scoring. Identification of proteins defining different regions (clusters) was performed by MLP scoring based comparison with bulk proteomics. The Fig in the inset of each panel shows part of volcano plot that marks respective positions of the identified proteins in the enrichment landscape. Scale bars—1 mm. GO term analysis was done based on Over-representation test that determines whether genes belonging to a specific GO term are present more than that would be expected (over-represented) in a subset of the experimental data. The yellow nodes in the network indicate the biological processes (named within the grey coloured ovals). The size of these nodes in the GO term maps (scale represented by the numbered black solid circles) are representative of the number of genes involved in each of them. The grey nodes along with their labels show the genes that are involved in the aforesaid pathways.

purposes. Experiments were further approved by the institutional animal care committee.

The AROM+ transgenic mouse was used as a systemic model of inflammation-associated male infertility developing with increasing age. This mouse line expresses the human androgen-converting enzyme P450 aromatase under control of a universal promoter (ubiquitin C) in an FVB/N background (Li et al, 2001; Li et al, 2003; Li et al, 2006).

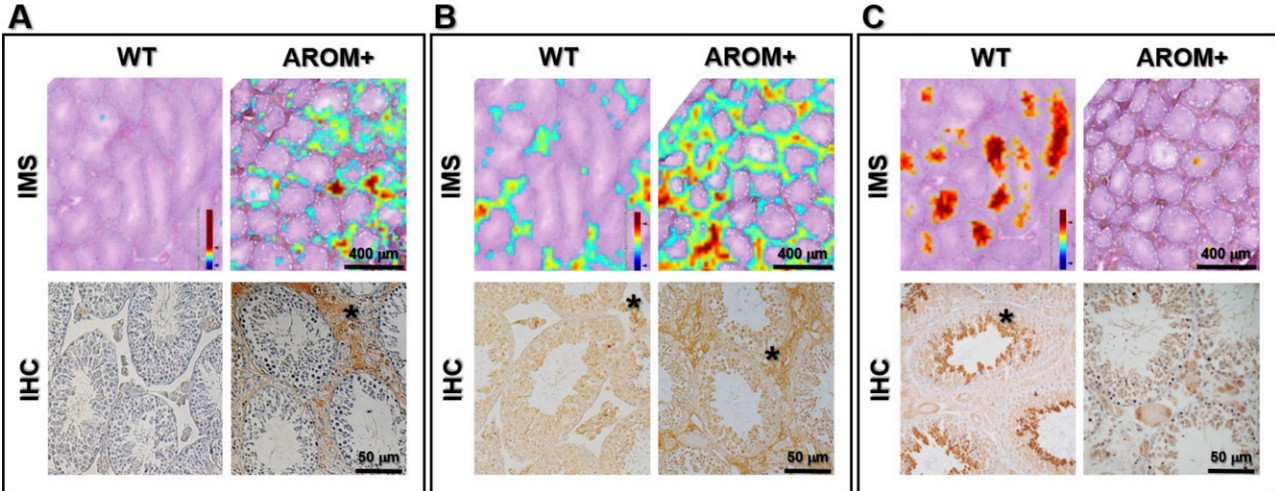

**Figure 6. Validation of MLP score based identification.**
Distribution pattern of some key proteins in imaging mass spectrometry (IMS) measurements (as identified by MLP scoring) and their corresponding corroboration by immunohistochemical staining (IHC). **(A)** Interstitial distribution of Mimecan (Ogn) (upper panel: ion image of m/z = 763.5 as measured in IMS; lower panel: IHC, interstitial staining indicated by *). In both IMS and IHC, the protein is observed to be present only in AROM+. **(B)** Interstitial distribution of Collagen I (Cola1) (upper panel: ion image of m/z = 929.5 as measured in IMS; lower panel: IHC, interstitial staining indicated by *). In both IMS and IHC, the protein is observed to be highly up-regulated in AROM+. **(C)** Tubular distribution of T-complex protein 1 subunit zeta (Cct6a) (upper panel: ion image of m/z = 784.5 as measured in IMS; lower panel: IHC, tubular staining indicated by *). In both IMS and IHC, the protein is observed to be present only in WT.

Testes were collected from ~320-d-old AROM+ mice (n = 3) and age-matching wild-type littermates (n = 3).

## LC–MS/MS measurements

### Sample preparation
Approximately 1 mg of tissue was incised from both WT and AROM+ mice testis. Initially the tissues were homogenized manually using Micro-homogenizers, PP (Carl Roth GmbH+Co. KG). Subsequently, proteins were extracted from the partially homogenized tissue using the iST Sample Preparation Kit (PreOmics) using manufacturer's protocol. Briefly; the tissues were lysed and subsequently sheared to get rid of DNA and other interfering molecules. The homogenous lysate was incubated with Trypsin at 37°C for 2 h 30 min. The ensuing peptides were subsequently desalted, purified and re-dissolved in 10 µl "Load" solution after drying completely in a speed vac.

### MS measurements
For reversed phase HPLC separation of peptides on a Ultimate 3000 nanoLC system (Thermo Fisher Scientific), 5 µl of the solution were loaded onto the analytical column (120 × 0.075 mm, in-house packed with ReprosilC18-AQ, 2.4 µm, Dr Maisch GmbH), washed for 5 min at 300 nl/min with 3% ACN containing 0.1% FA and subsequently separated applying a linear gradient from 3% ACN to 40% ACN over 50 min. Eluting peptides were ionized in a nanoESI source and on-line detected on a QExactive HF mass spectrometer (Thermo Fisher Scientific). The mass spectrometer was operated in a TOP10 method in positive ionization mode, detecting eluting peptide ions in the m/z range from 375 to 1,600 and performing MS/MS analysis of up to 10 precursor ions. Peptide ion masses were acquired at a resolution of 60,000 (at 200 m/z). High-energy collision-induced dissociation (HCD) MS/MS spectra were acquired at a resolution of 15,000 (at 200 m/z). All mass spectra were internally calibrated to lock masses from ambient siloxanes. Precursors were selected based on their intensity from all signals with a charge state from 2+ to 5+, isolated in a 2 m/z window and fragmented using a normalized collision energy of 27%. To prevent repeated fragmentation of the same peptide ion, dynamic exclusion was set to 20 s.

### Data analysis
Protein identification was performed by MaxQuant 1.6.0.16 software package. Parent ion and fragment mass tolerances were 8 ppm and 0.7 Da, respectively, and allowance for two missed cleavages was made. Mouse canonical protein database from UniProt (release June, 2018), filtered to retain only the reviewed entries was used for the searches. Regular MaxQuant conditions were the following: peptide FDR, 0.01; protein FDR, 0.01; minimum peptide length, 5; variable modifications, oxidation (M); acetyl (protein N-term); acetyl (K); dimethyl (KR); fixed modifications, carbamidomethyl (C); peptides for protein quantitation, razor and unique; minimum peptides, 2; minimum ratio count, 2. Proteins were validated on the basis of at least one unique peptide detected in the proteome of all the three replicates or in at least two of the three replicates.

### Statistical analysis
To detect significant changes in protein levels between WT and AROM+, we performed statistical analysis on the acquired LC–MS/MS data. We chose LIMMA moderated *t* test proposed earlier (Kammers et al, 2015) over standard *t* test because it uses full data to shrink the observed sample variance towards a pooled estimate, resulting in far more stable and powerful inference compared with ordinary *t* test.

Missing values in proteomics data is a common problem and are usually imputed before applying statistical methods. Here, we have used an imputation algorithm (Tyanova et al, 2016), which allows us to randomly draw values from a distribution meant to simulate

values below the detection limit of the MS instrument. We have implemented the statistical analyses in R, MATLAB and used Cytoscape (Shannon et al, 2003) for network graph generation.

### Pathway analysis

To understand biological pathways affected in testis by the over-expression of aromatase, we analysed proteins that were either enriched or exclusively detected in WT or AROM+ for their involvement in those pathways. Enrichment analysis is a common practice to discover biological themes associated with the proteins of interest. Here, we used the R package ReactomePA (Yu & He, 2016) to discover biological pathways in which the enriched or exclusive proteins participate. The package use hypergeometric distribution model to calculate P-values to determine whether any Reactome Pathways annotate a specified list of genes at a frequency greater than that would be expected by chance.

### IMS

### Sample preparation

Fresh frozen testis from WT and AROM+ animals (n = 2, in each case) were cryo-sectioned into 12 $\mu$m thick sections using CM1950 cryostat (Leica Microsystems) and thaw-mounted on conductive indium-tin-oxide coated glass slides (Bruker Daltonik GmbH) pre-treated with poly-lysine (1:1 in water with 0.1% NP-40). The slides were then stored at –80°C until further analysis.

Tissue sections were washed before trypsin and matrix spraying to remove lipids and salts, which act as interferents in MALDI measurements. Tissue sections were washed as previously described (Deutskens et al, 2011) with some modifications for measuring in situ generated peptides. Briefly; proteins were precipitated in situ by a sequential wash in 70% and 100% ethanol (for 30 s each) followed by a 90-s wash in Carnoy's fluid (6:3:1 ethanol:chloroform:acetic acid, vol/vol/v) to remove lipids and other interferants. Tissues were put back in 100% ethanol for 30 s to remove the excess chloroform. Further washing in dH$_2$O containing 0.2% TFA was performed to remove salts and acidify. Excess acid was removed by washing the sections in 20 mM ammonium bicarbonate for 30 s. The excess water from the previous two steps was removed by washing with 100% ethanol for 30 s. Subsequently, the slides were dried in vacuum overnight at RT before Trypsin-spraying.

25 ng/$\mu$l trypsin solution was prepared in 20 mM ammonium bicarbonate and sprayed on the tissue sections using the syringe spray system of TM Sprayer (HTX Imaging; HTX Technologies, LLC). The spraying parameters for trypsin were the following: temperature—30°C, no. of passes—8, pattern—criss-cross, flow rate—0.03 ml/min, nozzle velocity—750, track spacing—2 mm, gas pressure—10 psi, nozzle height—40 mm. The sprayed sections were then incubated at 50°C for 2 h 30 min in a humid chamber.

Subsequently, the slides were sprayed with 10 mg/ml HCCA in 70% ACN and 1% TFA in the TM sprayer using the following spray parameters: temperature—75°C, no. of passes—4, pattern—horizontal (HH), flow rate—0.12 ml/min, nozzle velocity—1,200, track spacing—3 mm, gas pressure—10 psi, gas flow rate—2, nozzle height—40 mm.

### IMS measurements

Imaging experiments were performed in a rapifleX MALDI Tissuetyper MALDI-TOF/TOF mass spectrometer (Bruker Daltonik GmbH) equipped with a SmartBeam 3G laser. Tissues were measured in a positive reflector mode with 25 $\mu$m spatial resolution and using a mass range of 600–3,200 Da. Each measurement was pre-calibrated externally using a commercial peptide calibrant mixture (Bruker Daltonik GmbH) spotted on the same target as the tissues at multiple positions. Tissues from all the animals were measured in the rapifleX using 1.25 GS/s sample rate and a pulsed ion extraction of 160 ns.

After the IMS measurements, HCCA matrix was removed by washing the slides in 70% ethanol and H&E counterstaining was performed. High-resolution images of the stained sections were obtained from Mirax Scan system (Carl Zeiss MicroImaging) and co-registered with MALDI-IMS data for histological correlation.

### Data analysis

The acquired data were exported to SCiLS Lab version 2016b (SCiLS, www.scils.de) (Trede et al, 2012) software for further analysis. Initially, a segmentation pipeline was executed on the acquired IMS data from both WT and AROM+ tissues (Fig S1) for unbiased hierarchical clustering with a minimal interval width of ±0.1 Da. The resulting clusters were examined for their correlation with histological features of the tissues manually. Subsequently, clusters representing relevant regions were selected for further analysis.

Peptide masses defined as distinguishing the clusters that are discussed in the article were found using "Find discriminative m/z values" tool of SCiLS Lab. The tool uses the Receiver Operating Characteristic measure to discriminate between all m/z values from a particular cluster. The resulting mass list in each case (with a threshold of at least 0.7) was exported as an excel file and further processed through in-house developed program (as described in the following section).

### Analysis and identification of peptide masses

Previous attempts aimed at identifying IMS peptides/proteins have resulted in either in very limited identification with considerable manual input or much improved identification with resource intensive setups and limited applicability (Alberts et al, 2017; Kriegsmann et al, 2017; Huber et al, 2018; Longuespée et al, 2019). The goal here was to identify proteins based on IMS peptides at the MS1 level with maximum likelihood. To achieve this, we performed multistage data analysis. The acquired IMS data were deisotoped taking advantage of co-localisation of peptides in different clusters. Deisotoping was performed on these co-localised peptides using standard tolerances (for m/z [±0.15] and for intensity [50%]). Each mass in the deisotoped IMS mass list was then matched with the LC–MS peptide masses (after adding the mass of a proton due to singly charged ions in MALDI) within a tolerance $\tau$. The choice of $\tau$ depends on the accuracy of IMS measurements and in this case $\tau = \pm 0.1$ was found to be appropriate (Fig S6 and Table S12). LIMMA statistics (as explained before) was applied on the LC–MS peptide data ("peptides" output file from MaxQuant analysis) and the peptides for the two groups WT and AROM+ were classified on the basis of fold change (log$_2$FC), defined as follows:

$$\log_2 FC = \log_2\left[\frac{\mu^{AROM+}}{\mu^{WT}}\right], \tag{1}$$

where $\mu^{AROM+}$ and $\mu^{WT}$ imply mean intensity of a particular peptide across the AROM+ and WT replicates, respectively. All peptides that

have $\log_2 FC > 0$ were classified as AROM+ and those that have $\log_2 FC < 0$ were classified as WT peptides. Subsequently, IMS masses were compared with either WT and AROM+ data depending on relative abundance of peptides in the tissues.

However, for one IMS peptide mass, the $\tau$-search method yielded multiple LC–MS peptides of similar masses (±0.1 Da) belonging to different proteins. Therefore, to determine the maximum likelihood of the peptide belonging to a particular protein, we devised a scoring system based on the following assumptions/conditions:

i. Peptides of relatively higher abundance are preferably detected in MALDI-TOF-IMS because there are no pre- or post-ionisation separation techniques involved. Therefore, it is more likely for a peptide detected in IMS to belong to an abundant protein than to its possible relatively less abundant counterpart.

ii. The search space for a peptide belonging to a cluster detected in IMS measurements would be restricted to either WT or AROM+ LC–MS data depending on the occurrence of the cluster in the respective tissues.

The aforesaid scoring system (most likely peptide: MLP) is then defined as follows:

$$MLP = \frac{\mu * \log_2 FC}{p}, \qquad (2)$$

where $\mu$ is the mean intensity of a particular peptide across three animals, $\log_2 FC$ is the fold change of that peptide for either WT or AROM+ and $p$ imply LIMMA moderated $P$-value of the same peptide. All these parameters were extracted from the LC–MS data of WT and AROM+ tissues. Therefore, higher the MLP score of an IMS peptide mass for a particular protein, more likely it is to belong to that protein. For all the IMS peptides, we then select the proteins with the highest MLP score, which are subsequently used to systematically investigate BPs and pathways to which they are associated.

We performed GO enrichment analysis by the means another R package clusterProfiler (Yu et al, 2012) to identify the enriched GO categories associated with proteins. This package uses hypergeometric distribution to calculate $P$-values to determine whether any GO terms annotate a specified list of genes at a frequency greater than that would be expected by chance.

### GO enrichment analysis

To find out the BPs and networks represented by the identified proteins in situ, we performed GO based enrichment analysis using overrepresentation test (Boyle et al, 2004) as implemented in clusterProfiler (Yu et al, 2012). In this, we assessed whether any GO-BP term annotates our list of proteins at a frequency greater than that would be expected by chance.

### Immunohistochemistry (IHC)

Additional samples of mouse testes were examined by IHC. They stem from previous studies. Testes were fixed in Bouin's fluid, embedded in paraffin (Li et al, 2006; Walenta et al, 2018), and sectioned to 5-$\mu$m thickness. Sections from 10-mo-old AROM+ mice (n = 3) and WT mice (n = 3) used for immunohistochemistry. IHC

staining was carried out as described previously (Mayer et al, 2016). For Mimecan, anti-mimecan (Ogn) antibody (G-1): sc-374463 from Santa Cruz (mouse monoclonal IgG$_1$) was used at a dilution of 1: 100. For Cct6a, anti-Cct6a (rabbit IgG) from Invitrogen (Cat. no. PA5-101926) was used at a dilution of 1:500. To stain the tissues for Collagen, affinity-purified, polyclonal rabbit anti-Col1 (R1038; Origene) antibody was used at 1:200 dilution (Welter et al, 2020). Negative controls consisted of omission of the primary antibody, of incubation with rabbit IgG or non-immune serum instead of the primary antiserum (Fig S7). Sections were counterstained with hematoxylin and visualized using a Zeiss Axiovert microscope equipped with a Jenoptik camera (ProgresGryphax Arktur; Jenoptik).

## Data/software availability

The codes for LIMMA statistics with data visualization, the $\tau$-search method, and MLP scoring modules will be available at the following GitHublink - https://github.com/wasimaftab/IMS_Shotgun. All proteomic data are available via ProteomeXchange with identifier pride:PXD022870.

## Supplementary Information

## Acknowledgements

The authors thank Dr Juan Antonio Aguilar-Pimentel from Helmholtz Zentrum Munich for providing the mice. Thanks are also due to the Group of Prof Dr Doris Mayr, Pathologisches Institut der (LMU München) Ludwig-Maximilians-Universität München, for their enormous help with cryo-sectioning of tissues for IMS measurements. We thank Dr Ignasi Forne of the Protein analysis unit (ZfP) at BioMedical Center (BMC), LMU, for his help with the LC–MS measurements. Thanks to Ms Irene Vetter, Molecular Biology department, Bio-Medical Center (BMC), LMU, for her excellent assistance in pre- and post-IMS measurement workflows. This work was supported in part by Deutsche Forschungsgemeinschaft (DFG) grants CRC1309 and CRC1123 (to A Imhof), MA1080/23-1; -2, 27-1, and 29-1 (to A Mayerhofer) and the DAAD/Academy of Finland (DAAD project 57347353; to A Mayerhofer, M Poutanen).

### Author Contributions

S Lahiri: conceptualization, formal analysis, supervision, validation, investigation, visualization, methodology, and writing—original draft, review, and editing.
W Aftab: software, investigation, in silico methodology, and visualization.
L Walenta: investigation and visualization.
L Strauss: resources and writing—review and editing.
M Poutanen: resources and writing—review and editing.
A Mayerhofer: conceptualization, formal analysis, funding acquisition, and investigation.
A Imhof: conceptualization, formal analysis, supervision, funding acquisition, investigation, project administration, and writing—original draft, review, and editing.

**Life Science Alliance**

## Conflict of Interest Statement

The authors declare that they have no conflict of interest.

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
