## [Reviewer comments · Life Science Alliance]

Life Science Alliance

MALDI-IMS combined with shotgun proteomics identify and localize new factors in male infertility

Shibojyoti Lahiri, Wasim Aftab, Lena Walenta, Leena Strauss, Matti Poutanen, Artur Mayerhofer, and Axel Imhof

DOI: <https://doi.org/10.26508/lsa.202000672>

Corresponding author(s): *Axel Imhof, Ludwig-Maximilians-University of Munich*

Review Timeline:

Submission Date:	2020-02-10
Editorial Decision:	2020-03-25
Revision Received:	2020-10-01
Editorial Decision:	2020-11-20
Revision Received:	2020-12-03
Editorial Decision:	2020-12-14
Revision Received:	2020-12-16
Accepted:	2020-12-16

Scientific Editor: Shachi Bhatt

Transaction Report:

March 25, 2020

Re: Life Science Alliance manuscript #LSA-2020-00672-T

Axel Imhof
Ludwig-Maximilians-University of Munich
Munich Center of Integrated Protein Science and BioMedical Centre
Großhadernerstr. 9
Planegg-Martinsried 82152
Germany

Dear Dr. Imhof,

Thank you for submitting your manuscript entitled "Imaging mass spectrometry and shotgun proteomics reveal dysregulated pathways in male infertility" to Life Science Alliance. The manuscript was assessed by expert reviewers, whose comments are appended to this letter.

As you will see, the reviewers provide constructive feedback on your work, and we would thus like to invite you to submit a revised version of your manuscript to us. Importantly, the value over/in addition to other approaches needs to get better fleshed out and the resource value of the dataset needs to get demonstrated via orthogonal approaches. Furthermore, reproducibility should get demonstrated and the source code and data need to be made available. Finally, it would be good to provide some level of new biological insight by following-up on a few candidates to enhance the impact of the study. I'd be happy to discuss this point further with you to see what could get added once the current pandemic allows taking-up lab work again (please see below that we do offer ample of time for revisions during the COVID-19/SARS-Co-V2 crisis).

The typical timeframe for revisions is three months. However, we are aware that many laboratories cannot function fully during the current COVID-19/SARS-CoV-2 epidemic and therefore encourage you to take the time necessary to revise the manuscript to the extent requested above. We will extend our 'scoping protection policy' to the full revision period required. If you do see another paper with related content published elsewhere, nonetheless contact me immediately so that we can discuss the best way to proceed.

Please note that papers are generally considered through only one revision cycle, so strong support from the referees on the revised version is needed for acceptance.

Thank you for this interesting contribution to Life Science Alliance. We are looking forward to receiving your revised manuscript.

Sincerely,

B. MANUSCRIPT ORGANIZATION AND FORMATTING:

*****IMPORTANT:** It is Life Science Alliance policy that if requested, original data images must be made available. Failure to provide original images upon request will result in unavoidable delays in publication. Please ensure that you have access to all original microscopy and blot data images

before submitting your revision.***

Reviewer #1 (Comments to the Authors (Required)):

The authors propose a novel method for identifying proteins in MALDI-IMS by combining information from bulk proteomics comparative analysis and MALDI-IMS. The authors illustrate the method by applying it to a mouse model of mouse male infertility.

The idea of combining a comparative analysis score (fold change or p-value) with the intensity from MALDI-IMS for enrichment analysis on MALDI-IMS data is novel and innovative. This indeed shall help with down-scoring peptides which are not differentially expressed and to clean up the enrichment analysis results. In my subjective opinion, this idea only makes the manuscript worth publication. However, the presentation can be improved and some claims, in particular "dysregulation of pathways" and that MALDI-IMS helped "revealing" it, may need to be toned down and formulated more strictly. For details, please see the minor and major issues formulated as follows:

MAJOR:

- 1) What is the new information that is provided by MALDI-IMS and was not captured by bulk proteomics? In the current presentation, the bulk results were largely omitted. I'm convinced that MALDI-IMS can and shall provide complementary insights. However, for the global proteome changes in testes (e.g. the low representation of the red and brown clusters from Fig 5 in the AROM+ mouse) -- they can likely be detected by bulk proteomics. Maybe there are some more subtle changes which are spatially-regulated that can only be detected by MALDI-IMS?
- 2) At the current point, there is no sufficient evidence provided to claim "dysregulation" of pathways. Basically, only Fig 5 presents the results on the MALDI-IMS comparative analysis. However, it only shows two segmentation maps, and then enrichment for clusters. There is no clear information whether these cluster do represent the difference between the conditions.
- 3) No results are shown about reproducibility of MALDI-IMS results. The images show only 1 WT and 1 AROM+ section (out of n=3 for each condition). Has the spatial segmentation delivered consistent clusters accross the replicates within every condition? How many technical replicates (sections) have you used for MALDI-IMS?
- 4) What is the biological meaning of the green cluster (Fig 5)? It was detected in both WT and AROM+ mouse (without clear difference between the conditions) and it's reported with marker proteins which are associated with immune response.
- 5) Please report the accuracy of the MALDI-IMS measurements at least for some key reported peptides, as rapiflex is known to suffer from low accuracy, potentially preventing using the +- 0.1 Da tolerance and requiring a higher tolerance. For validation and quality control, please show the m/z images of the reported peptides.
- 6) The GitHub repository reported in the paper is empty. For a paper proposing novel computational method, I'd require providing the source and data code before publication so that the reviewers can evaluate the code and reproduce the results. I'd also require providing both source code and data

to the future readers.

MINOR:

- 1) I'd suggest to refrain from using "likelihood" in the definition of your score, since "likelihood" has a particular definition in statistics. Your score does not necessarily satisfy it.
- 2) After you performed the scoring of peptides with the MLP score, you would still have ambiguous isobaric peptides. Have you evaluated how many of them were in the top list thus biasing the enrichment results? I can imagine that for a large set of isobaric peptides, this may produce false positives in the enrichment analysis.
- 3) Why only shared proteins were considered? Have you looked into proteins which were detected in the AROM+ mouse?
- 5) Fig 4B is confusing -- I'd expect the images corresponding to isotopic peaks of a peptide from the same cluster (e.g. brown peaks) to exhibit similar localization. What do grey noisy rectangles mean?
- 6) Fig 5:
 - What are the differences between the clusters shown in B and C? The enrichment results seem to be the same. So is there any difference?
 - What are the proteins/GO terms enriched in WT vs AROM+? It's not clear from this figure. I'd suggest to show the markers for each cluster on the volcano plot estimated from the bulk analysis or present results in some other way to show that the differences detected between WP and AROM+ are consistent between bulk and tissue analyses.

Reviewer #2 (Comments to the Authors (Required)):

In this study, Shibojyoti Lahiri et al. combined shotgun proteomics with MALDI imaging mass spectrometry to describe dysregulated pathways in a hormone-induced male infertility mouse model. To uncover the dysregulated pathways the authors used mice overexpressing human P450 aromatase that resulted in pathologically increased estrogen levels and compared it to WT mice. The infertility phenotype and the increased immuno active macrophages and aplasia in this transgenic mouse strain have been already described by Li X et al, 2001 and 2006. By using their newly developed data analysis pipeline the authors were able to provide a more detailed picture of the molecular processes affected.

Overall the paper is written in a clear, concise, and logical manner. While the technical aspect is a strength of this study, a weakness of the manuscript in its current state is the very limited biological insight it seems to provide on spermatogenesis biology. It remains to be elusive what this information means or how it affects the associated biology. The gathering of quantitative data might be of interest to the spermatogenesis community, but unfortunately, the authors do not provide any example of the usefulness of the information.

Specific comments

1. The title of the manuscript is very generic "...reveals dysregulated pathways..." and does not point towards the key findings of the study. Which dysregulated pathways?
2. Which altered pathway(s) is the cause and which one is the consequence of the AROM+ induced morphological changes in the testis? I think in order to better understand the effect of aberrant

P450 aromatase expression, the authors also have look at an earlier time point (not only at ~320 days old AROM+ mice) before the overall disruption of the tubular is fully established.

3. While MS imaging can simultaneously measure hundreds of specific molecular species, broad protein measurements remain a challenge due to issues related to ionization suppression and the absence of a separation dimension, even by using the authors' newly described 'maximum likelihood peptide score'. Laser-capture microdissection (LCM) provides an interesting alternative for collecting very small amounts of tissue since it can readily excise tissue regions as small as single cells. LCM has been successfully integrated into workflows for spatially resolved proteomics of low input tissue material (PMID: 28004771; PMID: 29491378; PMID: 29941660, <https://doi.org/10.1101/779009>). The ability to isolate different tissue compartments, and detect proteomic signatures, using as little as 5,000 cells significantly increase the prognostic and diagnostic ability (PMID: 31043742). Using LCM-based proteomics workflows has significantly advanced biomolecular science and is capable of identifying and quantifying several thousands of proteins. This approach would be a great counterpart to the here described bulk tissue proteomic analysis. Furthermore, the ability to use formalin-fixed, paraffin-embedded tissues for this type of analysis would also be exciting, as it negates the need to take a separate fresh sample for analysis, but instead, the same one obtained for traditional histology can be used. It also opens the possibility of using historical samples for research.

4. At least some proteins, which have been shown to be significantly down/up-regulated in the testis of AROM+ mice, should be confirmed by antibody-based histological methods (tissue microarray analysis).

Point by point response to the reviewers comments:**Reviewer #1:**

The authors propose a novel method for identifying proteins in MALDI-IMS by combining information from bulk proteomics comparative analysis and MALDI-IMS. The authors illustrate the method by applying it to a mouse model of mouse male infertility.

The idea of combining a comparative analysis score (fold change or p-value) with the intensity from MALDI-IMS for enrichment analysis on MALDI-IMS data is novel and innovative. This indeed shall help with down-scoring peptides which are not differentially expressed and to clean up the enrichment analysis results. In my subjective opinion, this idea only makes the manuscript worth publication. However, the presentation can be improved and some claims, in particular "dysregulation of pathways" and that MALDI-IMS helped "revealing" it, may need to be toned down and formulated more strictly. For details, please see the minor and major issues formulated as follows:

We thank the Reviewer for his/her appreciation of our work and its value for the field. In the revised manuscript we followed his/her suggestion and changed the title to: *"MALDI-IMS combined with shotgun proteomics identify and localize new factors in male infertility"* We find that the new title now represents our key findings better.

MAJOR:

1) What is the new information that is provided by MALDI-IMS and was not captured by bulk proteomics? In the current presentation, the bulk results were largely omitted. I'm convinced that MALDI-IMS can and shall provide complementary insights. However, for the global proteome changes in testes (e.g. the low representation of the red and brown clusters from Fig 5 in the AROM+ mouse) -- they can likely be detected by bulk proteomics. Maybe there are some more subtle changes which are spatially-regulated that can only be detected by MALDI-IMS?

We fully agree with the reviewer that MALDI-IMS provides additional and complementary insights not only into the factors involved in male infertility but also in the molecular composition of distinct cell populations under wildtype conditions. In fact, it is the intention of the manuscript to describe the benefit of using both techniques together. We hope that we managed to better illustrate this in the revised manuscript.

In the first part of our study we extensively used the bulk proteomic results to interpret the establishment of AROM+ phenotype and performed an elaborate pathway analysis based on the significantly enriched and exclusively detected proteins for each condition. This analysis pointed out multiple cellular processes that play important roles in the pathological state of hormone induced male infertility. These findings are displayed in figures 1 and 2 and discussed in the corresponding section of the manuscript.

In the second part of the manuscript we make use of the bulk analysis to identify peptides that distinguish certain regions of the tissue specifically by combining an unsupervised hierarchical clustering algorithm with our novel MLP scoring system.

The clustering method in itself reveals a much higher molecular complexity of the wildtype testis compared to tissue isolated from AROM+ mice. The MLP scoring now enables us to identify peptides that define specific tissue regions (Figure 5) and potentially establish the infertile phenotype. Many of these proteins like Vdac2, Clgn, Akap4, Dnajc3 distinguishing the red and brown cluster and Dnajb11 or Hspa5 distinguishing the blue cluster are proteins that are not enriched significantly in bulk analysis and would have been not considered for further investigation when only studying the bulk proteome. We have tried to clearly emphasize this aspect in the revised manuscript.

Another aspect of our study that we now also emphasize in the revised manuscript is a much better molecular description of the protein composition and a classification of the tubuli in wildtype testis. These peptide assignments should now enable us to define spermatogenic stages at the molecular level, which is currently a labour intensive and tedious effort.

2) At the current point, there is no sufficient evidence provided to claim "dysregulation" of pathways. Basically, only Fig 5 presents the results on the MALDI-IMS comparative analysis. However, it only shows two segmentation maps, and then enrichment for clusters. There is no clear information whether these cluster do represent the difference between the conditions.

We agree with the reviewer that using the term "dysregulation" might not be appropriate as of yet. However, the absence of most of the clusters observed in WT tissue in AROM+ clearly indicates a major difference in physiology (Figures 3 and 5). In order to differentiate explicitly, we have now included the LC-MS results in the comparison in Figure 5. This shows us that the proteins defining the clusters are actually enriched in the respective animals and a substantial loss or gain of those proteins involved in defining the regions are responsible for the establishment of the AROM+ phenotype.

Further, we have refrained from using the term "dysregulation" in the revised version of the manuscript. We have used more generic terms in order to explain our findings.

3) No results are shown about reproducibility of MALDI-IMS results. The images show only 1 WT and 1 AROM+ section (out of n=3 for each condition). Has the spatial segmentation delivered consistent clusters across the replicates within every condition? How many technical replicates (sections) have you used for MALDI-IMS?

We have measured multiple (n=3) tissue sections from different animal (n=3) for MALDI-IMS. Unbiased hierarchical clustering in ScilsLab could always distinguish between the WT and AROM+ tissues based on peptide measurements across all the sections measured. However, due to its inherent variability in the number of tubuli of a

particular stage the size, number and distribution pattern of clusters in different replicates can vary quite substantially.

As an example, we provide images of two biological replicates (figure below) in both conditions for the reviewer.

4) What is the biological meaning of the green cluster (Fig 5)? It was detected in both WT and AROM+ mouse (without clear difference between the conditions) and it's reported with marker proteins which are associated with immune response.

We thank the reviewer for pointing this out. As the green cluster is found under both conditions and as our MLP scoring algorithm relies largely on the differences between the two conditions, the identification of the marker peptides has to be taken with care. As such situations (clusters existing in healthy and diseased tissue) will very likely also occur with similar settings it is clearly a shortcoming of our method. We have now discussed this in the revised manuscript and suggest methods to overcome this issue.

5) Please report the accuracy of the MALDI-IMS measurements at least for some key reported peptides, as rapiflex is known to suffer from low accuracy, potentially preventing using the ± 0.1 Da tolerance and requiring a higher tolerance. For validation and quality control, please show the m/z images of the reported peptides.

According to the suggestion of the reviewer, we have now included the information about the accuracy of measurement and the corresponding m/z images of key peptides from both WT and AROM+ in the revised version of the manuscript (Figure S5 in the supplementary information). We observe that using an interval of ± 0.1 Da is sufficient for the current study. Increasing the interval does not provide us with any extra information or altered interpretation of the data. However, keeping in mind the point raised by the reviewer, we now allow variation of the mass accuracy interval in the bioinformatic pipeline. This will help the users to choose the interval according to their accuracy of measurement.

6) The GitHub repository reported in the paper is empty. For a paper proposing novel computational method, I'd require providing the source and data code before publication so that the reviewers can evaluate the code and reproduce the results. I'd also require providing both source code and data to the future readers.

We apologise to the reviewer for this apparent miscommunication. We have mentioned that the codes will be available at the GitHub repository post-publication. We are now providing all the codes and instructions on how to use them to reproduce the reported figures. All of these will be uploaded at the GitHub repository if and when the manuscript is accepted for publication.

MINOR:

1) I'd suggest to refrain from using "likelihood" in the definition of your score, since "likelihood" has a particular definition in statistics. Your score does not necessarily satisfy it.

We have replaced the term "likelihood" to avoid its confusion with the statistically defined meaning. We now use "likely" as a more colloquial term.

2) After you performed the scoring of peptides with the MLP score, you would still have ambiguous isobaric peptides. Have you evaluated how many of them were in the top list thus biasing the enrichment results? I can imagine that for a large set of isobaric peptides, this may produce false positives in the enrichment analysis.

We thank the reviewer for pointing out this important aspect of our work. We completely agree with the reviewer that presence of isobaric peptides will include potential false positives and subsequently bias the enrichment results.

However, we observe that our scoring approach effectively deals with this situation. As depicted in Figure 4 and mentioned in the results section, there is a possibility that multiple peptides (isobaric) from LC-MS measurements are matched to a single peptide from IMS due to relatively lower accuracy of the currently used IMS set up. The devised MLP scoring eliminates the isobaric peptides and identifies the peptide that is most likely to be the true one based on the assumptions mentioned in our Materials and

methods section. We would therefore assume that there are no isobaric peptides in the list that we use for our enrichment analysis. That our scoring system based on the assumptions are most likely true is now demonstrated by GO analysis corroborating the enrichment from LC-MS analysis and IHC validation.

3) Why only shared proteins were considered? Have you looked into proteins which were detected in the AROM+ mouse?

We apologize to the reviewer that we did not clearly point out the fact that we have considered both the shared and exclusively detected proteins for comparison. What probably led to this misunderstanding is the fact that we separated the analysis of shared proteins and the exclusive ones in order to avoid any analytical bias arising from imputation of missing values for the exclusive proteins in the system where they are undetected. In the manuscript, Figure 1 shows the analysis for shared proteins whereas Figure 2 shows the results of pathway analysis of the exclusively detected proteins.

4) Fig 4B is confusing -- I'd expect the images corresponding to isotopic peaks of a peptide from the same cluster (e.g. brown peaks) to exhibit similar localization. What do grey noisy rectangles mean?

We have tried to clarify the reviewer's concern in the revised version of the manuscript. The images corresponding to the isotopic peaks do show similar distribution. In order to make it clearer, we have now included the m/z images/data of the isotopic peaks in the supplementary information (Figure S4) of the revised manuscript.

The grey rectangles are graphical representations of datasets comprising of monoisotopic mass lists. We have now mentioned this explicitly in the figure legend in the revised version of our manuscript.

5) Fig 5:

- What are the differences between the clusters shown in B and C? The enrichment results seem to be the same. So is there any difference?

- What are the proteins/GO terms enriched in WT vs AROM+? It's not clear from this figure. I'd suggest to show the markers for each cluster on the volcano plot estimated from the bulk analysis or present results in some other way to show that the differences detected between WP and AROM+ are consistent between bulk and tissue analyses.

We agree with the reviewer that the enrichment results shown in Figure 5B and 5C appear to be similar. However, there are certain differences between the two clusters that can be comprehended upon closer inspection of the datasets (Supplementary files 9a and 9b). The red cluster in Figure 5B differs from the brown cluster in the presence of certain proteins like mitochondrial tricarboxylate transport protein, gametocyte specific factor 1, tudor domain containing protein, etc. These factors involved in the mitochondrial TCA cycle and male gamete formation are not present in the brown cluster. The brown cluster, on the other hand is characterised by proteins like Outer dense fiber protein 2, ADP-ribosylation factor-like protein 3, Leucine-rich repeat-containing protein 46, Tektin-5, etc that cumulatively enriches the cluster for processes related to cilium organization and cell projection assembly as compared to the red

cluster in Figure 5B. These findings are now explicitly mentioned in the Results section of our revised manuscript.

Based on the comment of the reviewer, we have now included a modified version of Figure 5 in the revised manuscript, where we have included part(s) of the volcano plot to demonstrate the consistency of our findings.

Reviewer #2:

In this study, Shibojyoti Lahiri et al. combined shotgun proteomics with MALDI imaging mass spectrometry to describe dysregulated pathways in a hormone-induced male infertility mouse model. To uncover the dysregulated pathways the authors used mice overexpressing human P450 aromatase that resulted in pathologically increased estrogen levels and compared it to WT mice. The infertility phenotype and the increased immuno active macrophages and aplasia in this transgenic mouse strain have been already described by Li X et al, 2001 and 2006. By using their newly developed data analysis pipeline the authors were able to provide a more detailed picture of the molecular processes affected.

We thank the Reviewer for his/her appreciation of our work and its value for the field. In the revised manuscript we tried to address all the concerns raised by his/her

Overall the paper is written in a clear, concise, and logical manner. While the technical aspect is a strength of this study, a weakness of the manuscript in its current state is the very limited biological insight it seems to provide on spermatogenesis biology. It remains to be elusive what this information means or how it affects the associated biology. The gathering of quantitative data might be of interest to the spermatogenesis community, but unfortunately, the authors do not provide any example of the usefulness of the information.

Specific comments

1. The title of the manuscript is very generic "...reveals dysregulated pathways..." and does not point towards the key findings of the study. Which dysregulated pathways?

We have now changed the title to better describe the key findings of the study (see answer to reviewer 1).

2. Which altered pathway(s) is the cause and which one is the consequence of the AROM+ induced morphological changes in the testis? I think in order to better understand the effect of aberrant P450 aromatase expression, the authors also have look at an earlier time point (not only at ~320 days old AROM+ mice) before the overall disruption of the tubular is fully established.

We agree with the reviewer that the time point that we look at in the current study represents stages near the phenotypic end point. Spermatogenesis is strongly affected here.

The main reason for choosing mice of this age is that they present a situation that is very close to human cases and therefore provide a strong phenocopy of the human

pathology (as described in Li et al., 2006). In addition, this stage provides a proof of principle of the developed method that leads to putative identification of proteins from the peptide profiles in IMS measurements. Investigating an earlier time point would definitely be insightful in understanding the effect. However, as this is quite resource intensive, we feel it is beyond the scope of our current manuscript.

3. While MS imaging can simultaneously measure hundreds of specific molecular species, broad protein measurements remain a challenge due to issues related to ionization suppression and the absence of a separation dimension, even by using the authors' newly described 'maximum likelihood peptide score'. Laser-capture microdissection (LCM) provides an interesting alternative for collecting very small amounts of tissue since it can readily excise tissue regions as small as single cells. LCM has been successfully integrated into workflows for spatially resolved proteomics of low input tissue material (PMID: 28004771; PMID: 29491378; PMID: 29941660, <https://doi.org/10.1101/779009>). The ability to isolate different tissue compartments, and detect proteomic signatures, using as little as 5,000 cells significantly increase the prognostic and diagnostic ability (PMID: 31043742). Using LCM-based proteomics workflows has significantly advanced biomolecular science and is capable of identifying and quantifying several thousands of proteins. This approach would be a great counterpart to the here described bulk tissue proteomic analysis. Furthermore, the ability to use formalin-fixed, paraffin-embedded tissues for this type of analysis would also be exciting, as it negates the need to take a separate fresh sample for analysis, but instead, the same one obtained for traditional histology can be used. It also opens the possibility of using historical samples for research.

We agree with the reviewer that LCM can indeed be a complementary method to IMS. However, compared to MALDI-IMS, LCM is quite labour and resource (time) intensive thereby preventing a systematic comparison of tissues from multiple patients and its use in a more clinical setting. We think that our workflow allows a more rapid screening of the molecular composition in biopsies of males suffering from idiopathic infertility once the peptides have been identified using our workflow. Moreover, the two methods are not mutually exclusive as MALDI-IMS can assist in defining regions of interest for LCM. We discuss the pros and cons of using LCM versus MALDI-IMS in the revised manuscript.

4. At least some proteins, which have been shown to be significantly down/up-regulated in the testis of AROM+ mice, should be confirmed by antibody-based histological methods (tissue microarray analysis).

According to the suggestion of the reviewer, we confirmed our claims by immunohistochemical staining of some key proteins in both WT and AROM+. The results have now been included in the revised version of the manuscript (Figure 6).

November 20, 2020

RE: Life Science Alliance Manuscript #LSA-2020-00672-TR

Dr. Axel Imhof
Ludwig-Maximilians-University of Munich
Munich Center of Integrated Protein Science and BioMedical Centre
Großhadernerstr. 9
Planegg-Martinsried 82152
Germany

Dear Dr. Imhof,

Thank you for submitting your revised manuscript entitled "MALDI-IMS combined with shotgun proteomics identify and localize new factors in male infertility". We would be happy to publish your paper in Life Science Alliance pending final revisions necessary to address reviewers' minor comments and meet our formatting guidelines.

As you will note from one of the reviewers' comments, they have requested source code data. I know that you shared the source code for the reviewers with us over email, and I did send them to the reviewer for their input, but I haven't heard back from the reviewer about it. Unfortunately, this wait also delayed our communication with you, but I have a feeling that my emails might have been lost in spam. Anyways, I request you to submit the source code along with the revised manuscript and I can re-send them to the reviewer for their final approval then.

Along with the points listed in the reviewers' comments and appended at the end of this email, please also attend to the following:

- please make sure that the author list in our system and in your manuscript match
- please upload your supplementary figures as singular files and add your supplementary figure legends to the main manuscript text (directly underneath your main figure legends)
- please use the [10 author names, et al.] format in your references (i.e. limit the author names to the first 10)
- please add a data availability section with accession information for large datasets (<https://www.life-science-alliance.org/manuscript-prep#datadepot>)
- would it be possible to label the supp. material files (or datasets) also as supplementary tables? and edit the callouts in the manuscript text accordingly?
- please provide scale bars for the bottom panels in Figure S4 A & B

To avoid unnecessary delays in the acceptance and publication of your paper, please read the

following information carefully.

A. FINAL FILES:

B. MANUSCRIPT ORGANIZATION AND FORMATTING:

Sincerely,

Shachi Bhatt, Ph.D.
Executive Editor
Life Science Alliance
<https://www.lsjournal.org/>
Tweet @SciBhatt @LSAJournal

Reviewer #1 (Comments to the Authors (Required)):

The authors have performed a good revision that improved the manuscript and the presentation of the results. However, a few issues are still left not addressed or they are addressed only to a limited degree:

Major:

3): I'd request including the results for all n=3 for each conditions in the manuscript or Supplementary Materials, including the segmentation maps as well as ion images for all peptides. They should be provided for all n=3 and not only for the reviewers so that the readers can judge the reproducibility.

5): Accuracy is not shown in Fig S5 which shows only ion images. An ion image by itself does not show the accuracy but rather the intensity values within the pre-specified m/z range selected for the ion image. Can you please show that your mass spec data has +/-0.1 Da accuracy over the whole imaged area, e.g. showing the m/z values of the centroided peak over the whole area.

6): Re the data and source code: as a reviewer, I'd like to see both the data and source code before the publication. For the data, I'd recommend submitting the data to a repository and provide a reviewer-only access. My request is also motivated by the general situation that in the field that often the data is not made available and this leads to reproducibility and re-use problems. I'm not sure what's the LSA requirements on this and would leave it up to the editor to decide on it.

Minor:

2): I don't understand how MLP scoring eliminates isobaric peptides. This is an important (although minor) aspect and should be presented carefully. Can you please explain it and present evidence?

Reviewer #2 (Comments to the Authors (Required)):

This revision has suitably addressed my concerns. The revisions provide increased clarity as well as a better discussion of the work in the context of previous studies in this area.

Reviewer #1 (Comments to the Authors (Required)):

The authors have performed a good revision that improved the manuscript and the presentation of the results. However, a few issues are still left not addressed or they are addressed only to a limited degree:

We thank the Reviewer for acknowledging and appreciating our revised manuscript. Below we provide responses to the specific comments.

Major:

3): I'd request including the results for all $n=3$ for each conditions in the manuscript or Supplementary Materials, including the segmentation maps as well as ion images for all peptides. They should be provided for all $n=3$ and not only for the reviewers so that the readers can judge the reproducibility.

As requested by the reviewer, we have now provided segmentation maps as a supplementary figure (Fig S1) for the readers and have referred at appropriate places in the main text. Although there was an overall broad difference between WT and AROM+ in all the experiments performed, there were 2 biological replicates where we could observe the finer structures as well. We have presented this and changed the information from $n = 3$ to $n = 2$ in the main text.

5): Accuracy is not shown in Fig S5 which shows only ion images. An ion image by itself does not show the accuracy but rather the intensity values within the pre-specified m/z range selected for the ion image. Can you please show that your mass spec data has ± 0.1 Da accuracy over the whole imaged area, e.g. showing the m/z values of the centroided peak over the whole area.

We have now included an estimate of the accuracy of several measured peaks in a new Supplementary file/table S11. Figure S6 is an overall indication that the interval choice of 0.1 Da does not include any additional peaks or leads to loss of information over the whole mass range leading to alteration of interpretation of the data. Now, in addition we have provided an estimation of the variation of those m/z values over the entire measurement areas (as calculated from 40 manually defined ROIs across the areas and corresponding to over 3000 individual spectra per condition) as a supplementary file/table. Further, as mentioned before, we would keep the choice of accuracy flexible, so that this does not limit the applicability of our method.

6): Re the data and source code: as a reviewer, I'd like to see both the data and source code before the publication. For the data, I'd recommend submitting the data to a repository and provide a reviewer-only access. My request is also motivated by the general situation that in the field that often the data is not made available and this leads to reproducibility and re-use problems. I'm not sure what's the LSA requirements on this and would leave it up to the editor to decide on it.

In the revised manuscript we have provided the following link for the used codes:

ftp://ftp.lrz.de/transfer/AROM_paper_codes/

The reviewer can access here all codes, corresponding files for all figures and a detailed instruction on how to run the codes. We will make the code publicly available as soon as the manuscript is accepted for publication.

Minor:

2): I don't understand how MLP scoring eliminates isobaric peptides. This is an important (although minor) aspect and should be presented carefully. Can you please explain it and present evidence?

The MLP scoring does not eliminate isobaric or near isobaric peptides but enables to do a correct assignement of the peptide to a given protein. To better explain this peptides we changed the phrase "To overcome this ambiguity" (ln 199,pg 10) to "To correctly assign such peptides to a particular protein". We hope this better explains the use of the MLP scoring system.

December 14, 2020

RE: Life Science Alliance Manuscript #LSA-2020-00672-TRR

Dr. Axel Imhof
Ludwig-Maximilians-University of Munich
Munich Center of Integrated Protein Science and BioMedical Centre
Großhadernerstr. 9
Planegg-Martinsried 82152
Germany

Dear Dr. Imhof,

Thank you for submitting your revised manuscript entitled "MALDI-IMS combined with shotgun proteomics identify and localize new factors in male infertility". We would be happy to publish your paper in Life Science Alliance pending minor revisions.

Thank you for providing the source codes and reviewer access, as you will see from the reviewer's comment below, the reviewer is now fully satisfied with this manuscript.

Thank you also for formatting the manuscript in accordance to the points mentioned in the previous email. There are a few discrepancies in the Supplemental figures that need to be rectified before we can accept this paper for publication,

- + Supplemental Figure 2 is cut off (was Supplemental figure 1 in the earlier version). Can you please correct it?
- + The resolution of Supplemental Figure 7 is very low. Could you please provide a higher resolution image for the same?

A. FINAL FILES:

-- High-resolution figure, supplementary figure and video files uploaded as individual files: See our detailed guidelines for preparing your production-ready images, <https://www.life-science->

alliance.org/authors

B. MANUSCRIPT ORGANIZATION AND FORMATTING:

Sincerely,

Shachi Bhatt, Ph.D.
Executive Editor
Life Science Alliance
<https://www.lsjournal.org/>
Tweet @SciBhatt @LSAJournal

Reviewer #1 (Comments to the Authors (Required)):

I thank the authors for addressing my requests and congratulate them on the performed work

December 16, 2020

RE: Life Science Alliance Manuscript #LSA-2020-00672-TRRR

Dr. Axel Imhof
Ludwig-Maximilians-University of Munich
Munich Center of Integrated Protein Science and BioMedical Centre
Großhadernerstr. 9
Planegg-Martinsried 82152
Germany

Dear Dr. Imhof,

Thank you for submitting your Research Article entitled "MALDI-IMS combined with shotgun proteomics identify and localize new factors in male infertility". It is a pleasure to let you know that your manuscript is now accepted for publication in Life Science Alliance. Congratulations on this interesting work.

DISTRIBUTION OF MATERIALS:

Again, congratulations on a very nice paper. I hope you found the review process to be constructive and are pleased with how the manuscript was handled editorially. We look forward to future exciting submissions from your lab.

Sincerely,

Shachi Bhatt, Ph.D.

Executive Editor

Life Science Alliance

<https://www.lsjournal.org/>
